



# Validation of StreamLine XR Doppler Lidar wind observations using in-situ measurements and WRF simulations

Tamir Tzadok[1], Ayala Ronen[2], Dorita Rostkier-Edelstein[2], Eyal Agassi[2], David Avisar[2],
Sigalit Bercovic[2], and Alon Manor[2]

[1]Life Science Research Israel (LSRI), Ness Ziona 7410001, Israel
[2]Environmental Physics Department, Israel Institute for Biological Research (IIBR), Ness Ziona 7410001, Israel

Correspondence to: Tamir Tzadok (tamirt@iibr.gov.il)

**Abstract.** Halo-Photonics Streamline XR doppler lidar measurements performed using several scan configurations (Velocity Azimuth Display - VAD and Doppler Beam Swing - DBS) and elevation angles of 60° and 80°, are compared to wind observations conducted by various in situ instruments (tethered balloon, meteorological mast, and radiosondes). Good agreement is obtained, with an $R^2$ over 0.90 for wind speed and a standard error <=18.90° for wind direction for the VAD scan method. Best performance was attained with VAD and lower elevation scans (60º). These results are consistent with the higher spatial lateral homogeneity exhibited by lower angle scans. The boundary layer structure along a diurnal cycle is analyzed by utilizing retrieved backscatter data and wind measurements in conjunction with WRF simulations. Presence of multiple inversions which allow the coexistence of different layers within the studied profile is also verified using data acquired by several radiosondes. Synergic use of Lidar data with WRF simulations for low SNR regions is demonstrated.

## 1 Introduction

The wind profile within the mixing layer is an essential parameter in various applications, such as weather forecasting, transport and aviation, and is a critical parameter in the wind energy industry. The most sophisticated and state-of-the-art technology available today for measuring the wind profile is a Doppler wind Lidar (Light Detection And Ranging). This instrument uses a pulsed laser beam and makes use of the principle of optical Doppler shift between the reference and backscattered signal obtained from the interaction between light and air aerosols. The shift in wavelength of the returned pulsed beam is used to determine the Doppler velocity in the line of sight (LOS) (Liu et al., 2019). Relative to older measuring techniques, such instruments promise some advantages, including large scan volume, mobility and 3-dimensional wind measurements, as well as relatively high temporal and spatial resolution. Thus, in recent decades, Doppler Lidars have been widely adopted in several real-life applications. For example, they are installed in airports to study aircraft-induced vortices and to detect wind shears. In the wind energy industry, they provide a promising alternative to in-situ techniques in wind energy assessment, turbine wake analysis, and turbine



control. Doppler Lidars have also been used in meteorological studies, such as observing boundary layers and tracking tropical cyclones, and can measure turbulence parameters (Bonin et al., 2017). Their ability to measure the backscatter signal makes them a tool for boundary layer measurements of fog (Ronen et al., 2021), clouds, etc.

Since Doppler wind Lidar provides only the wind component of the radial direction, and not the whole wind vector, some data processing based on mathematical manipulations is needed, to solve the whole wind vector out of the partial measured data. Various scanning methods are common for performing measurements with Doppler Lidars, especially the VAD (Velocity Azimuth Display) (Päschke et al., 2015) scan, a scan with a fixed elevation and a varying azimuth, and the DBS (Doppler Beam Swing) (Lane et al., 2013).

The use and the potential of Doppler Lidars in measuring winds are reported in various studies, including cross-comparison to other instruments, and validation of various scan modes. The Doppler wind Lidar we use is the Stream Line XR [Halo-Photonics, England. Distributed by METEK, Germany]. A few studies focusing on the correlation of StreamLine profile measurements with other measuring technologies, are summarized in Table 1. These studies used scan configurations with elevation angles up to 75° and averaging times in the range of 10-60 minutes.





**Table 1: References comparing wind measurement in StreamLine Wind Lidar and other instrumentations.**

| Reference | Scanning type [mode and azimuthal angle °] | Control system | Correlation values | Campaign period |
|---|---|---|---|---|
| Päschke et al., 2015 | VAD 75 | DWD 482 MGH Radar | RMSE = 0.62 Bias =0.2 | one year of data with 0.5-hour averaging |
| | | Rs92 Radiosondes [Vaisala] | RMSE=0.86 Bias=0.12 | |
| Lane et al., 2013 | DBS 75 | Sonic Anemometer [Gil instruments R3-50] | RMSE=1.12 Bias=0.81 | 3700 hours of data with 1 hour averaging |
| Mariani et al., 2020 | DBS 70 | Rs92 Radiosondes [Vaisala] | $R^2>0.81$ Bias=0.46 | 11 months of data |
| | VAD 70 | | $R^2>0.89$ Bias =0.27 | |


As is seen in Table 1, the correlation values are usually high. All studies agree that DBS and VAD scans provide a good estimation of wind speed and direction with a slight advantage to VAD scans measurements. The current research describes measurements designed to compare the StreamLine XR Doppler Lidar, with respect to other measuring methods, during a field campaign that took place in the northern area of Israel

during September, 2021. The novelty of the current study stems from the following elements:

1. All previous studies utilize a single measuring technology for cross-comparison of Lidar measurements. Every such technology has unique features and behaviour which affect measurement discrepancies. The campaign reported here involves the simultaneous operation of several independent technologies. Beside the Doppler Lidar itself, a tethered balloon, a multilevel 100 m



meteorological mast, and radio-sonde measurements were used. This significantly enhances the evaluation of the Lidar performance.

2. Past in-situ measurements for evaluation of Lidar performance of high altitudes was based solely on radio-sondes, which have limitations in terms of temporal and spatial continuity and coverage. Here, for the first time, continuous in-situ wind measurements, supported by a tethered balloon, are used

for the evaluation of high-altitude Lidar performance.

3. Extracting lateral wind components from Lidar measurements involves applying LOS with a certain elevation angle. Choosing the appropriate angle is important. On one hand, the smaller the angle (LOS closer to zenith), the more realistic is the assumption of lateral homogeneity. However, on the other hand, larger angles allow the radial component, directly measured by the instrument, to include

more of the lateral wind component. Here, this issue is addressed by applying all measurements using two distinct elevation angles of 60° and 80°.

4. The measurement technologies applied for the validation of the Lidar are limited in their spatial coverage. Here, the validation procedure is enhanced by utilizing high resolution wind and temperature fields, produced by the meso-scale numerical weather prediction model WRF, which

are compared to the Lidar measurements.

5. A synergetic approach for the simultaneous application of a Lidar measurements and WRF predictions is shown. The Lidar signal-to-noise ratio depends upon the concentration of aerosols in the boundary layer, which can be too low for altitudes above the capping inversion. Model predictions also involve intrinsic uncertainty. Here, studying the atmospheric profile using a

combination of the two allows for complementarity.

## 2 The field campaign: Location, Instrumentation and Models

Our field campaign took place in an open rural area near a fixed meteorological mast. The Lidar, tethered balloon, and radiosonde launching point were located 550 meters from the mast, and 29 meters lower than

its ground level, see site map in Fig.1. The site height is 88 meters above sea level, with an almost clear and uninterrupted path to the Mediterranean Sea from the northwest, and thus it is significantly affected by both synoptic and meso-scale events. According to reanalysis maps from ERA5 (Hersbach et al., 2020), during this campaign the synoptic pressure gradients were quite low, mostly forming a shallow Persian low (channel). This synoptic configuration is associated with weak synoptic winds, a shallower Marine inversion

and a larger dominance of sea and land breeze on wind direction and speed in the region study (Yair & Ziv, 2014).



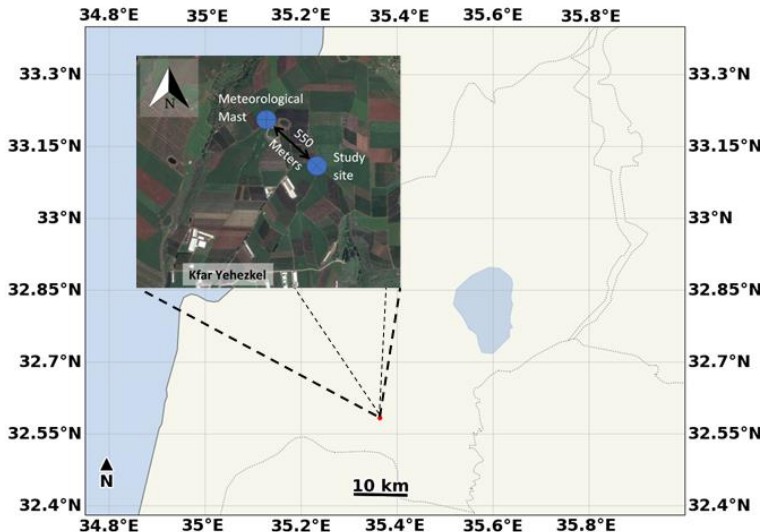

**Figure 1: A map of the campaign layout [© Google maps]. The Meteorological mast height is 115 meters above sea level (N 32° 35'11'', E 35° 21'36''). The Study site included the Streamline-XR and the balloon at (N 32°34'59'', E 35°21'51''), 550 meters south east to the mast, 88 meters above sea level.**

The 4 wind measurement instruments are shown in Fig.2 and they are described in the following figure. It should be noted that except for the Lidar, all instruments measure only the horizontal wind components.

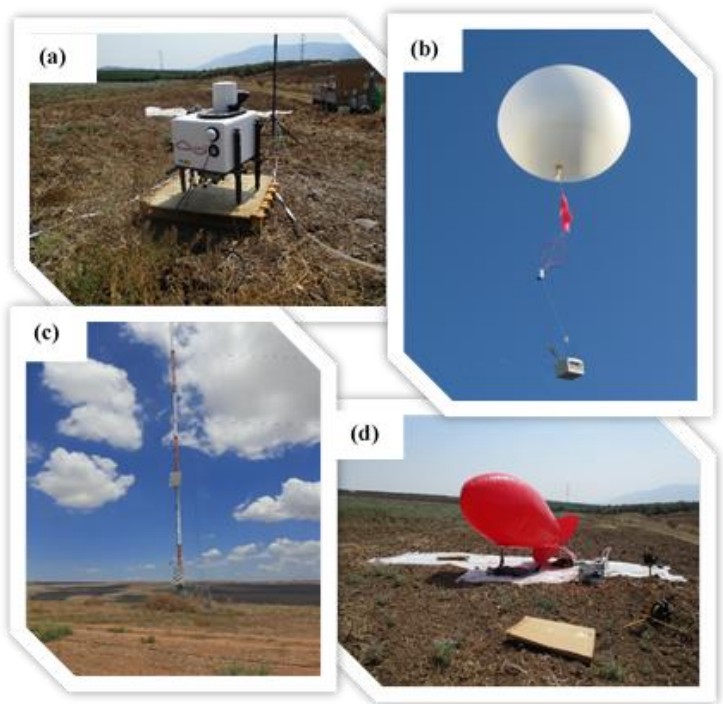


**Figure 2: Field campaign instrumentation. (a) The streamline-XR [Halo Photonics, England]. (b) radiosondes
[Modem, France]. (c) EDF meteorological mast at height 100 m. (d) A Zeppelin balloon [Vaisala, Finland].**

**2.1 Halo-Photonics Stream-Line XR Doppler Lidar**

The Stream-line XR, Fig.1(a), is a 100 MHz Doppler Lidar system with a pulse length of 310 ns associated
with a maximal resolution of 1.5 meters along LOS. The Stream-Line XR Doppler Lidar is a Doppler lidar
that has a full scan range, from the horizon to the zenith operated with mirrors. The maximum range of this
specific lidar due to physical and processing limitations is ~ 12 Km, though at large elevation angles the
range will be limited to ~1-3 Km (or the mixing layer height, MLH) due to insufficient aerosol concentration
aloft, which is needed for backscatter of the signal. Table 2 presents the physical attributes of this system.





**Table 2 – StreamLine XR Lidar technical specifications**

| Specification | |
|---|---|
| Laser Wavelength | 1550 nm |
| Laser Rap. Rate | 10KHz |
| Laser pulse length (FWHM) | 310 ns |
| Output power (average) | 400 mW |
| Average beam opening | 60 urad |
| Scanner angels from zenith (degree) | Elevation (-13 - +193) |
| Scanner angle resolution (mRad) | 0.2 |
| Scanner angular speed (mRad/sec) | 1020 |
| Wind velocity range (m/s) | 38 |
| Wind velocity precision (m/s) | 0.04 |
| Minimal range for measurements (m) | 30-40 |
| Maximal range for measurements (m) | 6000-12000 |
| Range resolution (m) | 1.5 |
| System Dimensions in m (WxLxH) | 0.6 X 0.5 X 0.4 |

**2.2 Tethered Balloon**

The Vaisala Zeppelin system, see Fig.2(b), consists of 5 sondes mounted to the cable anchoring the Zeppelin. The sondes measure wind velocity and direction and also measure altitude, pressure and temperature with a random sampling time between 10 sec to 1 min per each sonde. During this study the sondes were mounted between 150 meters to 900 meters in increments of about 100 m. Data availability was partial due to communication problems, hence the acquired data included 2-5 sondes at a time.


**2.3 Meteorological Mast**

The meteorological mast, shown in Fig.2(c), is owned by "EDF Renewables Israel" and "Blue Sky Energy" and operated by "NextCom". 2 cup anemometers measure horizontal wind speed at heights of 60 and 80 meters above the ground (89m and 109 m above the Lidar's location) and wind direction is measured by 2

vane sensors at 45 m and 75 m (74m and 104m above Lidar's location). For the campaign purposes, 10 min averaged data were used.



**2.4 Radiosondes**

18 Radiosondes [M10 Meteomodem, France], see Fig.2(d), were launched during the campaign and provided the meteorological parameters of the atmosphere up to altitudes of ~ 30 Km for about an hour and a half. The sampling frequency is 1 sec and the balloon vertical speed is 4-6 m/sec.  A typical radiosonde trajectory is seen in Appendix A in Fig.A2.

**2.5 WRF model**

The WRF model is one of the most advanced mesoscale numerical weather prediction models available, designed to serve both operational forecasting and atmospheric research needs. Prognostic variables include horizontal and vertical wind components, various microphysical quantities, potential temperature, geopotential, and surface pressure. WRF model has a nesting grid capability that allows zooming into a sub-region with high horizontal resolution by generating a series of higher resolution child grids within the coarser

parent grids. WRF includes a complete suite of physics schemes that accounts for the important atmospheric and land-surface physics. Several different formulations are available for each of these schemes, used to define the model topography and other static surface fields. For a complete description of the WRF modeling system, see, e.g., Skamarock et al. (Skamarock et al., 2021).

We use the WRF model (version 4.0) with the Advanced Research WRF (ARW) solver for the simulations.

We ran a two-way 3-nested domains configuration. The three domains with horizontal grid sizes of 13.5, 4.5 and 1.5 Km are shown in Fig. A1. The model configuration including vertical grid, physical parameterizations, numerical options and initialization time was chosen as in the reference runs in (Avisar et al., 2021). The ERA5 global reanalysis (Hersbach et al., 2020) was used for initial and boundary conditions.

**3 Methods**

3.1 Comparison of Doppler lidar relative to other measurement techniques

The Doppler lidar StreamLine XR was operated in three scan methods. In order to allow flexibility, the scan modes presented in this work were three procedures that we wrote by us (and not the supplied, built-in scan schemes). These procedures were a "stare" scan in the vertical direction, a "VAD" scan with 24 beams and

a "DBS" scan with 3 beams: vertical LOS, north LOS with a fixed elevation and east LOS with a fixed elevation. To study the sensitivity to elevation angle, scans were operated in two elevation angles of 60° and 80°. The Lidar was configured to operate with a gate length of 12 points (which results in a corresponding gate length of 18 meters) with an overlapping sequence that produced a measurement spatial resolution of 1.5 m in LOS. Above the mixing layer, the returned signal diminishes quickly due to the rapid decline of

aerosol concentration in the atmosphere. The in-situ measurements from other instrumentation were at altitudes under 1000 m; for this reason and as a consequence of the lidar's limitations in measuring in low



aerosol density, the number of gates was limited to 3840 with an overlapping 18-meter gate length, resulting in a maximum altitude measurement of 5764m. The signal-to-noise ratio (SNR + 1) threshold (for valid retrieved values) advised by the manufacturer is ~1.015 in order to reduce the error in measuring the wind

velocity. In this study however, the value was set to 1.007, in order to increase data availability, as presented in (Päschke et al., 2015).

A scanning operation procedure was built to perform six different scans automatically. This program was operated continuously with a cycle time of ~ 5 min, as follows:

- A DBS scan at 60º with 3 beams and a duration of 14 sec. (referred to as "DBS 60").
- A VAD scan at 60º with 24 beams and a duration of 36 sec. (referred to as "VAD 60").
- A DBS scan at 80º with 3 beams and a duration of 14 sec (referred to as "DBS 80").
- A VAD scan at 80º with 24 beams and a duration of 36 sec (referred to as "VAD 80").
- A VAD scan at 75º with 6 beams and a duration of 6 sec.
- A stare scan at 90º and processing with a duration of ~ 1:30 min.

Extraction of the wind components $u$, $v$, and $w$ out of the Lidar series of LOS measurements was done similarly to the methods presented in (Päschke et al., 2015; Lane et al., 2013) (Our methods and equations are presented in Appendix B). The extraction is performed under the assumption that the winds are uniform along each horizontal cross section of the scan volume. In order to get an estimation of wind uniformity, each

wind speed measurement was compared to a sinusoidal wave (see the method presented by Smalikho and Banakh, 2017), where high correlation indicates wind uniformity. It is important to note that this check is performed with the assumptions that the vertical component contribution is negligible.

During this study we focused on the verification of the horizontal wind speed and direction. The vertical component was not included out because all the other instruments only measure the horizontal wind

component.

Since the assembly of the measurement devices differ in their sampling rates, an averaging time interval is introduced in order to have a common time base for comparison. A 30-minute averaging time interval was selected, which compromises an adequate number of samples in each interval with the requirement of flow persistence.

Heights of lidar data for comparison were selected to optimally match the location of the different in-situ sensors. This is not trivial for the tethered balloon sensors, which do not, in general, keep a constant altitude. Ascent and descent of the balloon takes time throughout which the height of the sensors keeps changing. Furthermore, even in its full rise position, the balloon drifts in the wind direction, which affects the orientation of the cable and alters the heights of the sensors. To enable comparison to lidar data, the heights of each

sensor were averaged over each 30 minutes time period to yield a single representative height. For cases in which the absolute median deviation of a sensor height throughout the averaging period was larger than 15 m, the measurements for that period were disregarded.





The agreement of Lidar observations with each of the corresponding instruments was assessed differently for wind speed and for wind direction. In the latter case, the agreement was statistically expressed by calculating

the average difference (Bias)-

$$Bias = \frac{1}{N}\sum_{i=1}^{N} d_i \,,$$  (1)

its standard deviation (STD)

$$STD = \sqrt{\frac{1}{N}\sum_{i=1}^{N}(d_i - BIAS)^2}$$  (2)

and the Root Mean Square Error (RMSE)

$$RMSE = \sqrt{\frac{1}{N}\sum_{i=1}^{N} d_i{}^2} \,,$$  (3)

where $d_i = L_i - C_i$ is the mutual difference for time period $i$ between the wind direction measured by the lidar, $L_i$ and measured by the corresponding instrument, $C_i$. Circular periodicity is taken into account by altering $d_i$ according to

$$d_i = \begin{cases} d_i - 360 & d_i > 180 \\ d_i + 360 & d_i < -180 \end{cases} .$$  (4)

For wind speed, agreement was statistically estimated by goodness of fit (R-squared) of linear regression.

3.2 Meteorological feature analysis of the boundary layer structure

Capping inversion layers limit aerosol vertical advection which affects the distribution of aerosol concentration along the boundary layer profile. Indirect indication of inversion may also be manifested by the existence of sharp wind speed and direction gradients. Thus, observed attenuated backscatter and wind

profiles may be utilized to determine the location and extent of elevated inversion layers.

Here, the ability of identifying important features of the boundary layer structure using Lidar observations in conjunction with WRF simulations is manifested for a typical diurnal cycle of a typical single day. The profiles are also validated by direct in-situ temperature profile observations acquired by five radiosondes launched throughout the considered time range.

**4 Results**

**4.1 Statistical cross-validation of Lidar observations**

Comparison of the Lidar observations to the tethered balloon, mast, and free radiosondes was done separately for each of the scan modes VAD 60, VAD 80, DBS 60, DBS 80. Fig.3(a) shows, as an example, a scatter plot comparing wind speed observations between the Lidar VAD 60 scan (shown in the following to perform

optimally) and the tethered balloon. A dashed red curve corresponds to an ideal agreement between the two data sets, which is closely matched (R-squared 0.98) by a linear best fit (equation embedded in the figure), shown by a solid blue curve.



Fig. 3(b) presents the corresponding comparison for wind direction. The angular extent of each arc corresponds to the average difference of wind directions, and its center is in the average wind direction. To

depict sensitivity to wind speeds, the calculation is done separately for different wind speeds, shown by the radial position of each arc. Bias and STD values (eq. 1 and eq.2) are also presented.

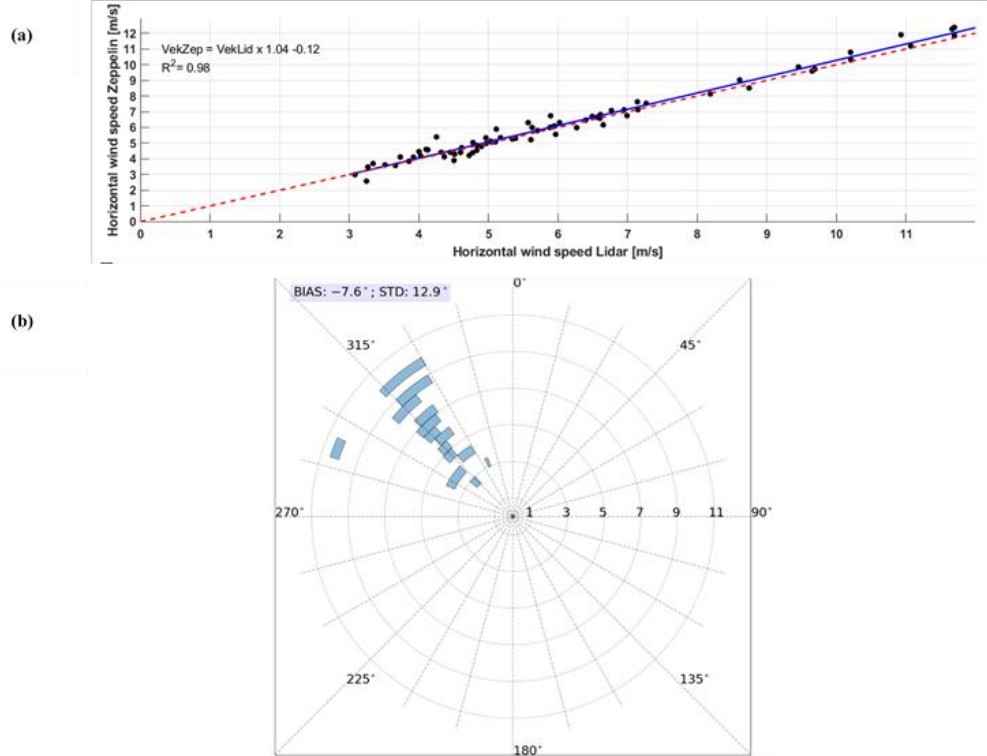

**Figure 3: correlation of wind speed and direction (from 30-minute averaged data, 13.9.2021-15.9.2021) between Lidar VAD 60 scans and Zeppelin measurements. (a) – Linear scatter plot of horizontal wind speeds. (b) -Average**
**wind direction difference between Lidar and tethered balloon measurements for VAD 60 scans. The difference is represented by the arc angular extent and the corresponding wind speed by its radial position.**

A similar comparison between the Lidar VAD 60 scan and the mast sensors is presented in Figure 4. Good agreement in the horizontal wind speed is observed, with $R^2 = 0.94$ and a bias of -0.12 m/s. The wind direction
standard deviation is 13.3° and the bias is -1.6°, with a higher difference in the wind direction at lower horizontal wind speeds, a trend that is seen in all the scans and mast comparisons.

A summary of the comparison between all Lidar measurements versus tethered balloon and mast observations
is given in Table 2. Regardless of the scan mode, horizontal wind speed agreement between lidar and balloon observations is good, with $R^2 >= 0.92$ and bias $<= 0.33$ m/s, respectively. STD for the wind direction is $<=18.9°$ and the absolute bias$<=7.6°$.



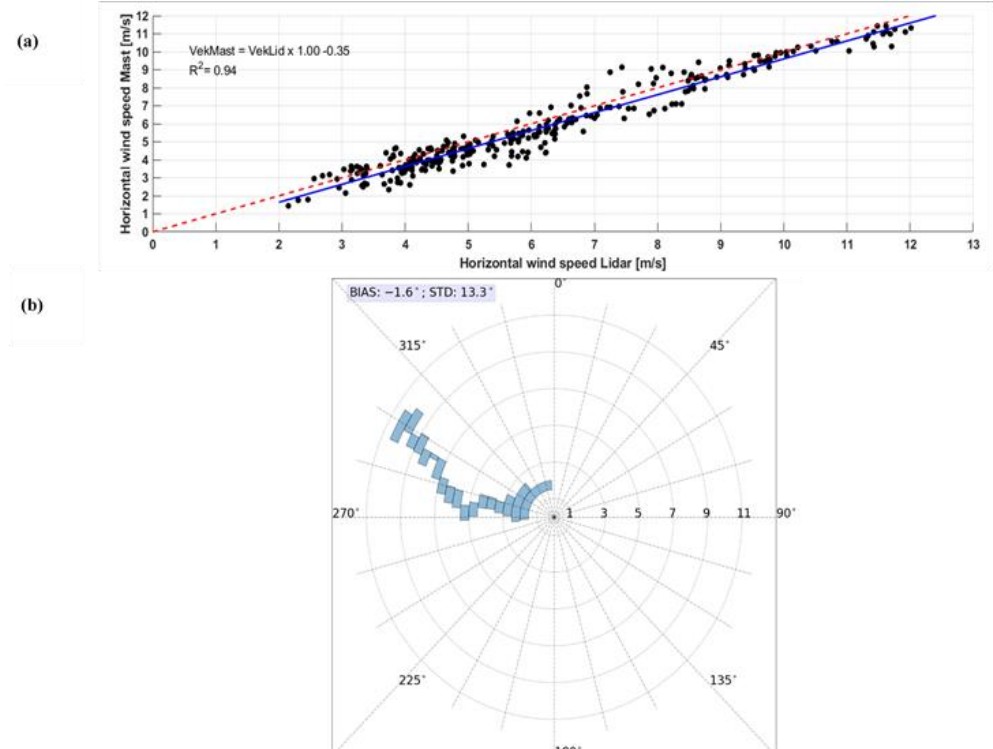

**Figure 4: correlation of wind speed and direction (from 30-minute averaged data, 13.9.2021-15.9.2021) between**
**Lidar VAD 60 scans and mast measurements. (a) – Linear scatter plot of horizontal wind speeds. (b) -Average**
**wind direction difference between Lidar and mast measurements for VAD 60 scans. The difference is represented**
**by the arc angular extent and the corresponding wind speed by its radial position.**

The agreement of Lidar and mast observations is also good with $R^2 >= 0.90$ and an absolute bias $<= 0.45$ m/s
for wind speed. The wind direction STD is $<=16.8°$ and the absolute bias is $<=2.8°$. For both the mast and
balloon comparisons, 60° elevation scans perform slightly better than 80° scans, and VAD scans seem slightly
superior to DBS scans. Overall, VAD 60 scans exhibit the highest agreement.





**Table 2- Summary of comparison between Lidar scans and other instrumentations**

|  |  | VAD 80 | VAD 60 | DBS 80 | DBS 60 |
|---|---|---|---|---|---|
| Lidar VS balloon | Correlation of horizontal wind speed | | | | |
|  |  | $R^2 = 0.92$ $Bias = -0.14$ $Rmse = 0.66$ | $R^2 = 0.98$ $Bias = -0.12$ $Rmse = 0.35$ | $R^2 = 0.92$ $Bias = 0.33$ $Rmse = 0.68$ | $\boldsymbol{R^2 = 0.94}$ $\boldsymbol{Bias = 0.19}$ $\boldsymbol{Rmse = 0.57}$ |
|  | Correlation of horizontal wind direction | | | | |
|  |  | $Bias = -5.9$ $SD = 15.2$ | $Bias = -7.6$ $SD = 12.9$ | $Bias = -6.6$ $SD = 18.9$ | $\boldsymbol{Bias = -7.1}$ $\boldsymbol{SD = 14.6}$ |
| Lidar VS mast | Correlation of horizontal wind speed | | | | |
|  |  | $R^2 = 0.90$ $Bias = -0.45$ $Rmse = 0.76$ | $R^2 = 0.94$ $Bias = -0.35$ $Rmse = 0.57$ | $R^2 = 0.90$ $Bias = -0.06$ $Rmse = 0.76$ | $\boldsymbol{R^2 = 0.92}$ $\boldsymbol{Bias = -0.28}$ $\boldsymbol{Rmse = 0.68}$ |
|  | Correlation of horizontal wind direction | | | | |
|  |  | $Bias = -2.8$ $SD = 13.9$ | $Bias = -1.6$ $SD = 13.3$ | $Bias = 0$ $SD = 16.8$ | $\boldsymbol{Bias = -1.4}$ $\boldsymbol{SD = 13.9}$ |

**4.2 Spatial uniformity**

It is of interest to quantitatively explore the origin of difference between scans with 60 and 80 elevation angles. This can be done by assessing wind uniformity along horizontal cross sections within the cone-shaped volume formed by VAD scans. Quantifying uniformity can be done (Päschke et al., 2015) by applying sine wave fitting. Consider a VAD scan with $V_i$ the radial velocity measurement $i$ for the azimuthal direction $\theta_i$. A least squares best fit sine function $v_i = Asin(\theta_i + B) + C$ is calculated, and the measure for wind uniformity is thus estimated by the R-square goodness of fit, i.e., $R^2 = 1 - \frac{\sum_i(V_i - v_i)^2}{\sum_i(V_i - \overline{V_i})^2}$. which, for perfect uniformity, equals unity.


Fig 5. shows R-square values of sine-wave fitting for VAD 60 (a) and VAD 80 (b) scans. It is shown that a scan at a 60º angle has more areas in the profile with higher $R^2$ values, compared to a scan at an 80º angle.

This means that a 60º scan better meets the wind uniformity assumption upon which wind component extraction is based. This conclusion is consistent with the above observation of improved performance for VAD scanning at 60º over corresponding scans with 80º elevation angle.

A note regarding the counter intuitiveness of this result should be made. One could have expected that higher elevation angles, which reduce the scanning volume, would improve spatial homogeneity yielding higher $R^2$

values. However, a side effect of narrowing the cone-shaped scan volume is the enhanced introduction of vertical velocity into the measured radial velocity. Unlike the lateral velocity which spatially fluctuates by no more than a few tens of percent (and much less for strong winds), the vertical wind can fluctuate drastically and even change its direction (upward\downward) between different locations along the scan cross section. Therefore, introducing more of the vertical component actually degrades spatial homogeneity, leading to

lower $R^2$ values for the 80º angle. As expected, this is especially pronounced for the morning and early afternoon hours (Fig. 5), for which boundary layer convective instability promotes enhanced vertical turbulence.

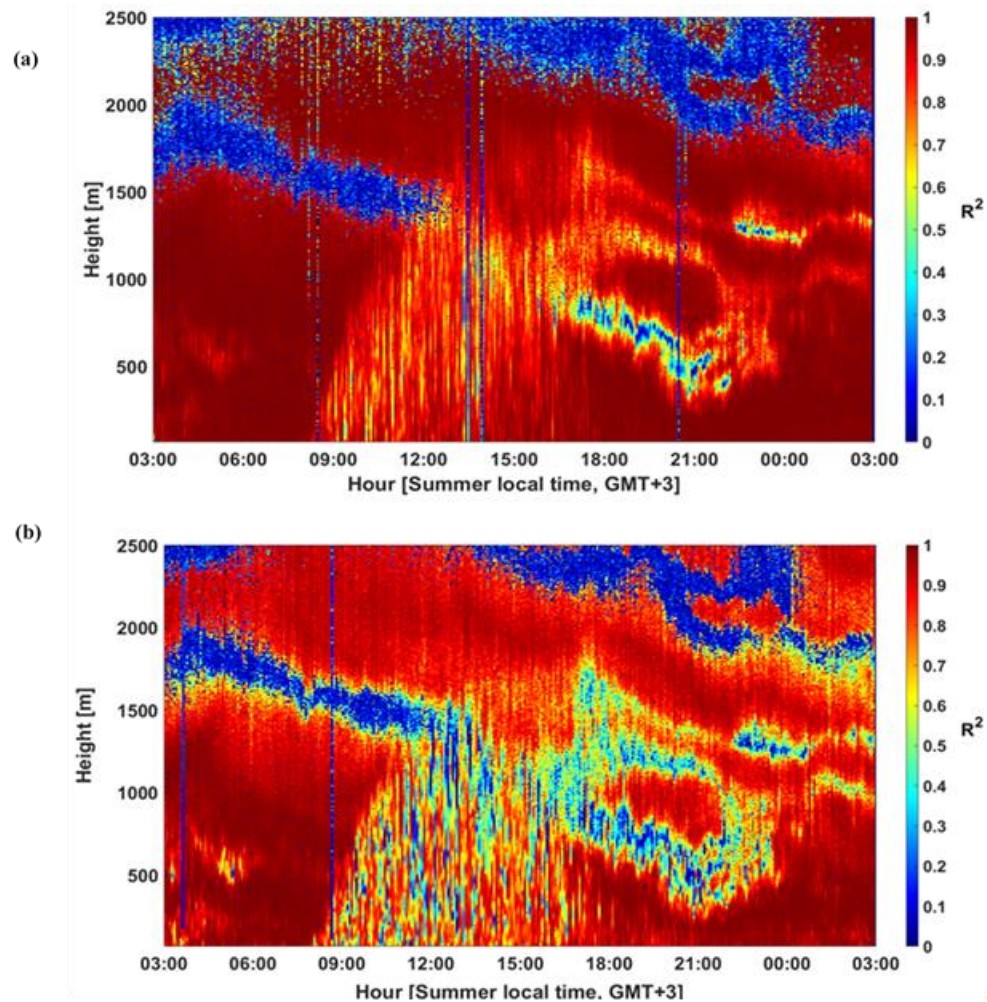

**Figure 5: Wind uniformity for the 14/9/20 as expressed by correlation between LOS doppler speed measurements and sine wave function. (a) – for VAD 60. (b) - for VAD 80.**





### 4.3 Boundary layer structure analysis

Aside from direct wind observations, Lidar measurements may serve to analyze the boundary layer structure,
including the indirect identification of temperature capping inversion layers, which decouple the profile into
different layers characterized by different flow regimes. In the following section, the synergic use of Lidar
observation and WRF simulations for the analysis of boundary layer structure and features will be shown for
a specific diurnal period.

An indication regarding boundary layer structure can be given by attenuated backscatter profiles. Fig. 6 shows
profiles of the attenuated backscatter ( $\beta$ [$\frac{1}{sr \times m}$] ) for the day 15-9-20. High values correspond to higher
concentrations of scattering elements in the atmosphere such as aerosols and hydrosols. The profiles change
throughout the period, exhibiting layers of high and low concentrations. Persistence of concentration layering
may suggest the existence of strong inversion layers, inhibiting vertical motion and decoupling adjacent
layers.

Five radiosondes were launched in the period of interest, measuring temperature, pressure, dew temperature,
and relative humidity. Elevation ranges along the profile which are characterized by positive temperature
gradients monotonically persisting along more than 50 meters are identified as inversion layers and are
marked by black contours in Fig. 6. Evidently, temperature inversion layers correspond to gradients in aerosol
concentration. The identification of capping inversion layers is thus evident in backscatter Lidar observations.


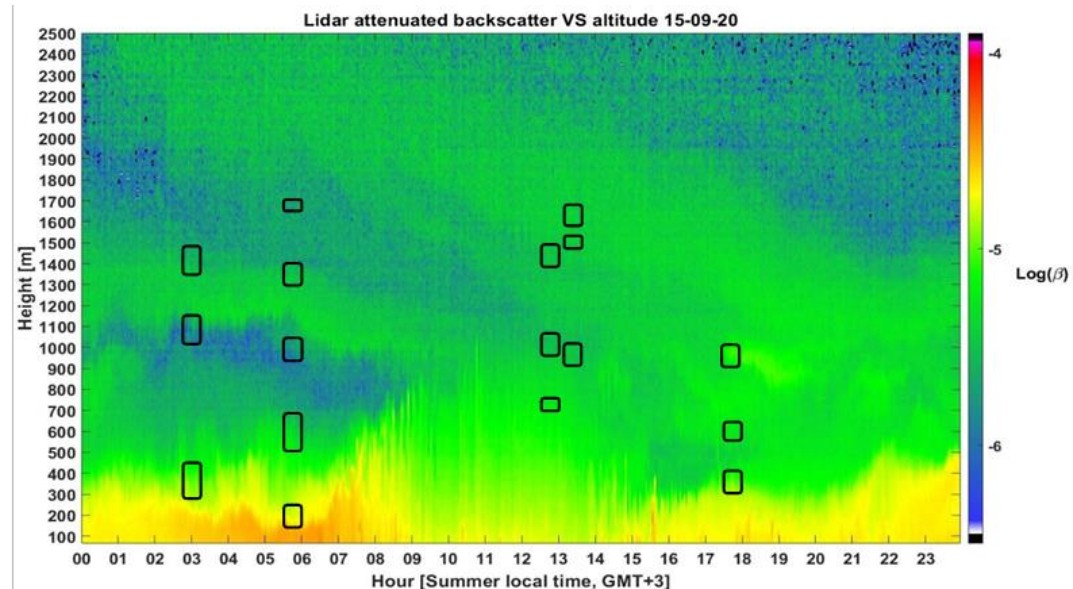

**Figure 6:  Lidar log attenuated backscatter ($Sr^{-1}m^{-1}$ ) from a "stare" mode scan, SNR+1 limited to 1.007. The
black boxes represent inversion in temperature from Radiosonde data**





In areas where inversion layers inhibit vertical airflow, momentum fluxes are reduced, and, consequently, strong and local wind speed and direction gradients may be observed along the profile. Fig.7 presents that phenomenon by showing the horizontal wind profile measured by the Lidar. The direction of the arrow depicts the horizontal wind direction (arrow pointing down corresponds to northerly wind, etc.), and their length and color corresponds to wind speed. Inversion layers, as identified by radiosonde observations, are

shown by gray shading. It can be seen that in a large number of cases the inversion layers are associated with wind speed or direction gradients.

By comparing Fig.6 and fig.7, a better understanding of the profile structure is obtained. The profile can be divided into 3 layers: 1. A low altitude layer with western northerly winds which persist throughout the diurnal period. 2. An intermediate layer characterized by very weak winds which changes in thickness

throughout the day. 3. An upper layer with strong south western to south eastern winds. A significantly impaired part of the Lidar data can be seen at the upper right part of Fig.7, shown as randomly directional weak winds, starting at around 13:00 at 2500 m altitude, and gradually thickening to a layer of ~1400- 2500 meters at the end of the day. As Fig. 6 reveals, this missing part of the graph is caused by a low attenuated backscattering signal, leading to low SNR values. Interestingly, the same range of heights and times can be

recognized in WRF simulations (Fig. 11) as containing exceptionally low mixing ratio values.

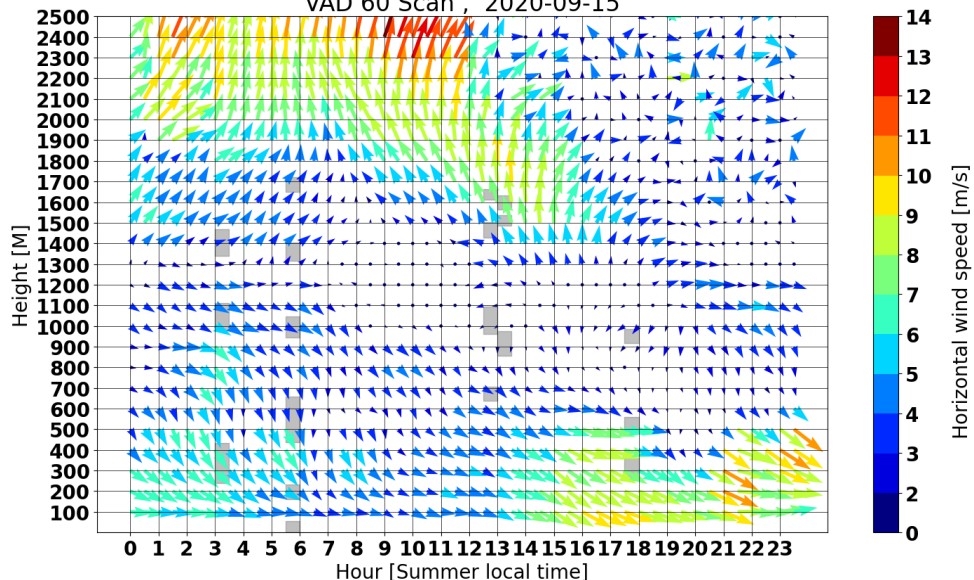

**Figure 7: Mean horizontal wind profile in 30-minutes increments out of Lidar measurements from VAD 60 scans. The arrows indicate the wind direction, while their color shows the wind speed. Each horizontal wind represents a mean of a 30-meter layer around the altitude presented. The grey boxes presented are inversion layers with**
**thickness >50m extracted from radiosonde temperature data.**

Fig. 8 presents the corresponding horizontal wind profiles that were produced by a WRF model, with the shaded blocks marking the inversion layers. Features of the boundary layer are well obtained, including the





division to 3 layers. The good agreement between the profiles observed by the lidar and the simulated profiles
may serve to support utilization of the modeled wind in areas where the Lidar SNR is too low to obtain valid
observations. This refers to the upper right part of Fig 7, for which low SNR leads to invalid data. The model
simulations (Fig. 8) for these parts can fill in the missing information and allow a more complete analysis of
the boundary layer diurnal dynamics.

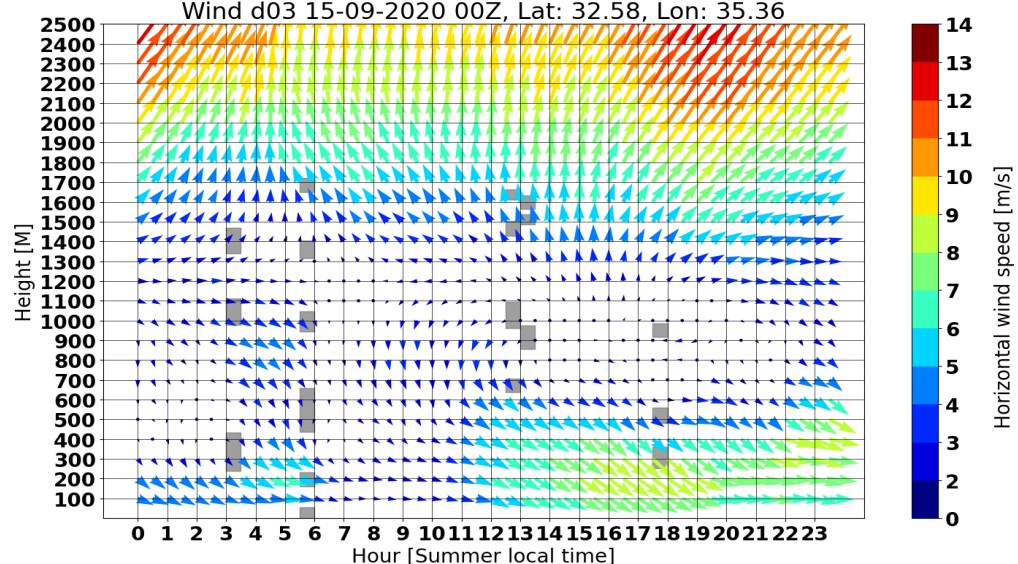

**Figure 8: Wind profile from WRF model, an average of 30 minutes interpolated for 100-meter gaps. The arrows
indicate the wind direction, while their color shows the wind speed. The black boxes presented are inversion layers
with thickness >50m extracted from radiosonde temperature data.**

The synergic use of Lidar observation with WRF simulations can be further supported by the analysis of
radiosonde data. Fig.9 and Fig.10 present radiosonde measurements, lidar wind profile, and the
corresponding WRF simulations during 5:30-6:00 and 17:00-18:00, respectively, representing conditions of
large SNR (in the morning) and poor SNR (afternoon) for altitudes above 1500 m. For the first case, good
agreement between wind profiles (Fig. 9, left panel) is evident. For the afternoon case, (Fig 10. a) wind
profiles obtained by the lidar and by radiosonde exhibit good agreement except for above 1500 m. In this
range, for which the Lidar data is invalid, WRF satisfactorily replicates the wind measured by the radiosonde,
although with a slight overestimation of wind speed.

Radiosonde observations can be further utilized to validate the analysis of the diurnal boundary layer
structure. Radiosonde temperature and specific humidity profiles (Figs. 9-10, two rightmost panels) reveal
the locations and extent of multiple inversion layers (shaded in the figure) which decouple adjacent layers.
For the early morning case (Fig. 9) an inversion layer around 600 m buffers a near surface layer with high
moisture content, and a dryer layer extending up to 1000 m. Two additional inversion layers allow the
formation of another high humidity layer between 1000 m and 1300 m. The suppression of momentum





exchange associated with inversion layers leads to the expression of these layers though the wind profile (left panel). The boundary layer is divided into layers with north-westerly winds (ground to 1000 m), western winds (1000 m - 1300 m), and south westerly winds aloft.

The evening radiosonde observations (Fig. 10) can be similarly analyzed. Here, two low inversion layers at 300 m and 500 m, and a subsequent one at about 1000 m, divide the boundary layer into a very moist shallow layer up to 300 m, followed by a much dryer layer up to 500 m, another layer between 600 and 1000 m, where the humidity increases, and a fourth layer aloft. The first two lower layers are associated with the sea breeze and characterized by strong westerly winds blowing from the sea. Fig. 11 shows that the high value of specific humidity and the typical flow in this layer starts at the late morning hours and lasts until after sunset. A sharp inversion between them enables their coexistence with very different water vapour mixing ratios (5 g/Kg in the upper and 16 g/Kg in the lower). The wind direction in the intermediate layer ranging between 600 m and 1000 m is easterly, which identifies it as the return current, associated with the sea breeze phenomena. The wind above the last inversion layer is south westerly which can be identified as the synoptic flow.

It is thus shown how WRF predictions may be used in conjunction with Lidar measurements to detect boundary layer features, to allow a more complete analysis of its structure, and to mitigate poor SNR conditions by completing Lidar observation with a WRF prediction.

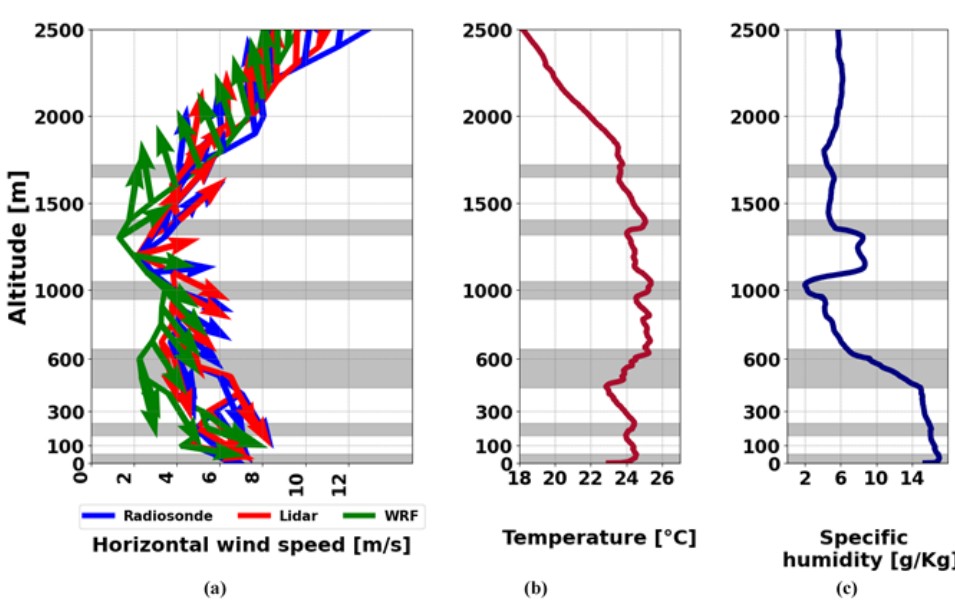

**Figure 9: Comparison of Lidar, WRF averaged data from 5:30-6:00 and radiosonde data from 5:53 launch, 15/09/2020. (a) Radiosonde wind measurement comparison with lidar VAD 60 wind measurement. (b) radiosonde temperature (c) radiosonde specific humidity. The gray layers presented are inversion layers with thickness >50m extracted from radiosonde temperature data.**

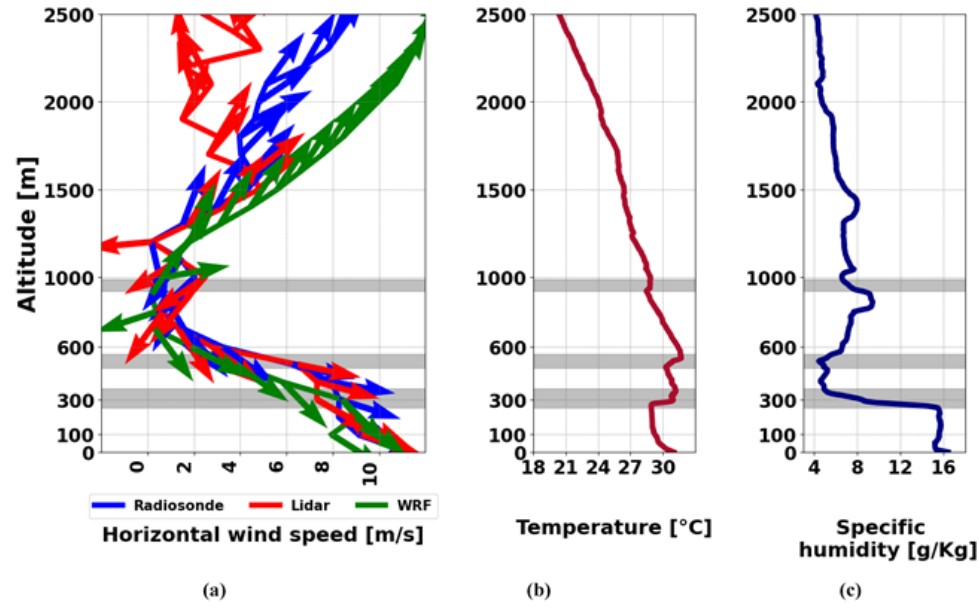


**Figure 10: Comparison of Lidar, WRF averaged data from 17:30-18:00 and radiosonde data from 17:57 launch, 15/09/2020. (a) Radiosonde wind measurement comparison with lidar VAD 60 wind measurement. (b) radiosonde temperature (c) radiosonde specific humidity. The gray layers presented are inversion layers with thickness >50m extracted from radiosonde temperature data.**

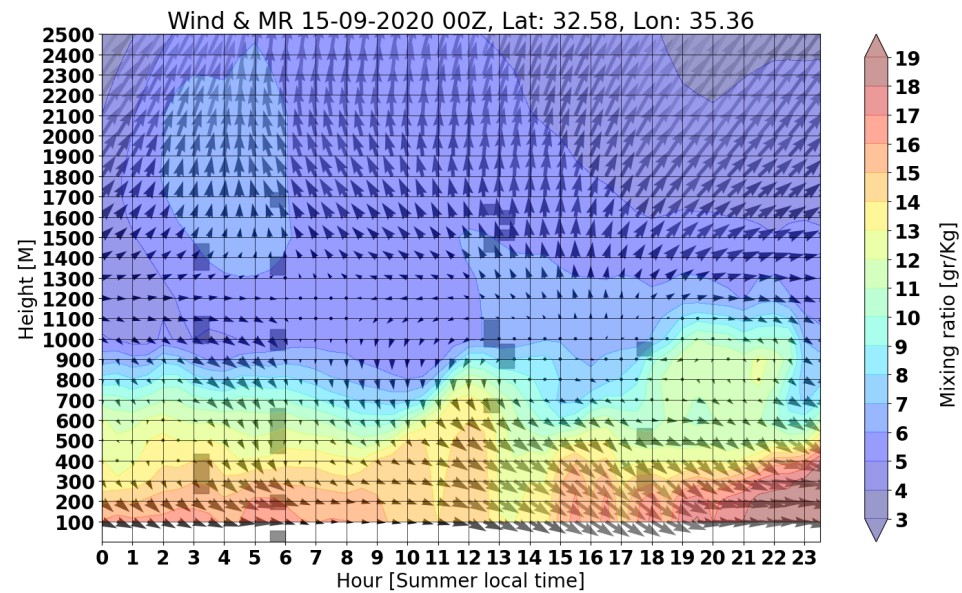


**Figure 11: Mixing ratio diurnal profile extracted from WRF runs, dark rectangles presented are inversion layers with thickness >50m extracted from radiosonde temperature data.**



**5 Discussion and Conclusions**

Data acquired by a Doppler wind lidar include two distinct parts of information. The first, and most essential
product is the wind component along the LOS, from which the full three dimensional wind vector can be
extracted using various scanning techniques and optimization algorithms. The second part is the attenuated
backscatter signal, which is usually used to evaluate the quality of wind measurement but, as manifested here,
can be used to infer the complete boundary layer structure. In this study, measurements of the Halo Photonics
Streamline-XR Doppler lidar are evaluated for both the wind and the backscatter signal, each using a different
approach.

Wind observations of the Doppler lidar were compared to several in-situ wind measurement instruments,
which included a meteorological mast, a tethered balloon, and free radiosondes. All comparisons showed
very good agreement with high $R^2$ values. Performance sensitivity to different scan methods (DBS and VAD)
and different elevation angles (60 and 80) is addressed.

Using the backscatter signal to study the whole boundary layer structure is manifested by combining WRF
model simulations and radiosonde observations, both yielding profiles of temperature, humidity, and wind.
Model data and radiosonde observations show good agreement, and reveal multiple inversion layers. Due to
capping effect of inversion layers, which influences the vertical distribution of aerosol concentration, these
inversion layers are clearly expressed by the lidar backscatter measurements.

Out key findings are:

- VAD scans slightly outperformed DBS results.
- Although spanning a wider area with potential spatial variability, scans with 60° elevation of the
  lidar are advantageous in comparison with 80° scans. This stems from the decrease of vertical wind
  component presence in the measured signal, combined with the fact that the vertical component
exhibits enhanced spatial variability, especially in daytime conditions.
- Reliable information regarding the existence of capping inversions can be obtained through the
  analysis of backscatter signal.
- A synergic use of WRF simulations and lidar observations is utilized for their mutual verification,
  specifically for low SNR regions, and for the understanding of the meso-scale processes governing
the dynamics of the boundary layer structure.
- Data from standard regional weather prediction models forecasting temperature and mixing ratio
  profiles can be used for pre-assessment of SNR as a function of height.

**Acknowledgements**

The authors would like to thank "EDF Renewables Israel" and "Blue Sky Energy" companies for their
generous help in supporting the meteorological data in "Kfar Yehezkel" station.
The authors would like to thank Yehuda Alexander for his review and helpful comments.





**Appendix A:**

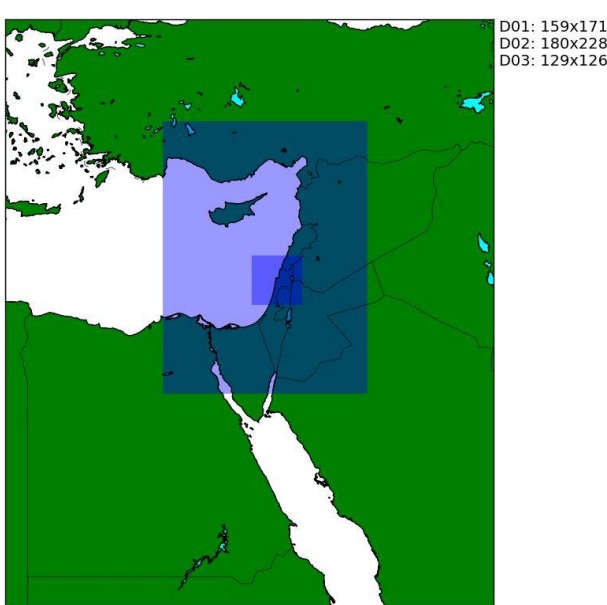

**Figure A1: The WRF model three nested domains: the outer domain with 159X171 grid cells at a resolution at 13.5 km, the intermediate domain with 180X228 grid cells at a resolution of 4.5 km and the inner domain with 129X126 grid cells at a resolution of 1.5 km.**


**Figure A2: a map of radiosonde trajectory until 2500 meters in altitude. (a) launch 5:53 15/09/20 (b) launch 17:57 15/09 20.**





**Appendix B:**

The methods of extracting u, v and w wind components from beam measurements from each scan.

Extraction of u,v and w form VAD scan set of measurements is done by solving n equations with 3 variables.

$$Vd(\theta_n, \Phi) = \begin{cases} u \cdot sin(90 - \Phi) \cdot cos(90 - \theta_n) + v \cdot sin(90 - \Phi) \cdot sin(90 - \theta_n) \\ +w \cdot cos(90 - \theta_n) \end{cases} \quad \text{(B1)}$$

u, v and w are the wind components, $\Phi$ is the elevation angle of measured the beam, $\theta$ is the azimuth angle

of the measured beam and n is the beam number in the scan (In this campaign 24 beams were used in each

VAD scan resulting in 24 equations with 3 variables). To solve the overdetermined equation set, a least

squares method was applied by the "lsqr" function in Matlab.

DBS scan assumes that the measured beam upwards (to the zenith) $Vd_Z$ represents well the w component

of the wind. Extraction of u and v was done by solving these 3 equations.

$$Vd_N(\Phi) = v \cdot sin(90 - \Phi) + w \cdot cos(90 - \Phi) \quad \text{(B2)}$$

$$Vd_E(\Phi) = u \cdot sin(90 - \Phi) + w \cdot cos(90 - \Phi) \quad \text{(B3)}$$

$$Vd_Z = w \quad \text{(B4)}$$

u, v and w are the wind components, $\Phi$ is the elevation angle of the measured beam. $Vd_N(\Phi)$ is the measured

doppler speed along the beam in the North direction at an elevation angle $\Phi$.

$Vd_E(\Phi)$ is the measured doppler speed along the beam in the East direction at an elevation angle $\Phi$.



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
