# Peer review of "Validation of StreamLine XR Doppler Lidar wind observations using in-situ measurements and WRF simulations"

_Atmospheric Measurement Techniques, 2022_

## Author Comment (AC1)

General: The author compared Halo doppler lidar using several scanning configurations with several in-situ wind measurement instruments, discussed analysis of boundary layer structure using retrieved Halo backscatter signals and wind measurements in conjunction with WRF simulations and demonstrated restoration of Halo wind data with low SNR regions combined with WRF simulations. While this is a moderately interesting piece of work, it contains nothing particularly new.  This paper is significantly lacking in a number of areas in including writing, and scientific content.  The paper can be acceptable with major revisions.

1. Author should discuss uncertainties for these instruments (tethered balloon, meteorological mast, and radiosondes).

**Response:**
These are discussed now throughout chapter 2, along with the description of the instruments.

2. The author should consider the influence of weather conditions on lidar wind retrieval with different elevation angles.

**Response:**
The campaign took place in calm weather conditions. The effect of boundary layer stability, which changes during the day, on the wind uniformity assumption is manifested and discussed in the current revision in l. 301.

3. I think a 30-minute averaging time is too long and will remove some features of wind speeds and direction.

**Response:**

We do agree that shorter averaging periods, of the order of 10 minutes, are preferable. However, in this work, a relatively wide range of scan configurations (methods and angles) was examined, which resulted in lower data availability specific to each method. A choice of time averaging shorter than 30 minutes did not yield sufficient data per period to attain meaningful averaging.

4. How many wind profiles are used to get Tab.2?.

**Response:**

For the meteorological mast

96 average profiles when each average has 9-12 lidar profiles, this resulted in 192 points for comparison.

The tethered balloon data went through a few filtering stages:

low speed winds filtering (<2m/s), filtering of large variation of heights during the averaged period. Furthermore there was lower availability of data due to maintenance issues.This resulted in 74 points (each point is an average of 9-12 lidar profiles) for comparison.

This is embedded in table 3 in the revision.

5.  I am concerned about the fig.9 and 10.

5.1 Aerosol density above 2000 m from 5:30 and 6:00 is higher than from 17:30 and 18:00. However, lidar background noise is low at night. Combing them, author should show SNR of Halo to prove it.

**Response:**

The SNR profiles corresponding to the two refered time ranges are plotted in the revision (fig.8). Indeed, lower SNR values are evident for the time range 17:30-18:00, which corresponds to the discrepancy between lidar and radiosonde observations.

5.2 Aerosol density between 1500m and 2000 m from 5:30 and 6:00 is lower than from 17:30 and 18:00. However, the wind difference of lidar and  radiosonde between 1500m and 2000 m from 5:30 and 6:00 is still better than from 17:30 and 18:00. Can author explain it and what cause it?

**Response:**

The major decline of snr and attenuated backscatter values for the evening radiosonde launch starts at above 1800 m (fig.6 and fig.8 in the revised ms) . This corresponds to the sharp deviation in wind speed and direction between Lidar and radiosonde observations (fig.11 b,c in the revised ms).

---

## Author Comment (AC2)

I have some issues with the novelty of the manuscript. As pointed out by the authors in the introduction, a major selling point of the manuscript is that the validation here has a wider range of reference instruments compared to previous validation exercises. However, on the other hand the manuscript only considers one site and a narrow range of atmospheric conditions. Therefore, I am not fully convinced that the manuscript achieves its goal to provide a more general validation.

**Response:**
Operating a tethered balloon system is an expensive and demanding task. This inherently limits the typical time period for a campaign of the kind reported here. In this respect, the novelty of the work is also the source of its weakness. This is explicitly discussed in the new revision (l. 67-70). Having said that, we believe that within the wide and variable literature on the subject, a validation against simple in-situ instruments, which so far has not been reported, is justified.

Another aspect of the manuscript – the inclusion of WRF data – is interesting, but as it is, it does not go anywhere in my opinion. The supporting role of WRF in the interpretation of the lidar measurements could have been substituted with existing literature and reanalysis of global weather models.

**Response:**
WRF is a regional meso-scale model. As such, It enables greater spatial resolution compared to global models. This is specifically true for the vertical, for which the global models poorly resolves the mixing layer with a typical layer thickness of 100-200 meters, while for the WRF model a few tens of layers are located within the mixing layer. Therefore, the good agreement of inversion layer heights, presented in this study cannot, in general, be attained with a global model. Likewise, some of the supportive role WRF forecasts can have for comprehending Lidar observations may refer to local, low level, boundary layer phenomenon. An example in the current study is the height and time of formation of a low level jet formed just after 20:00 (fig 7 and 9). This was noted in the revised version (l. 369)

The introduction should include more relevant literature and it could be more precise. The methods  are too sparse with information on the instruments to which the Doppler lidar is compared. The manuscript would also benefit from further polishing (consistent formatting, correct notation for units, etc.).

**Response:**
- More relevant references were added to the introduction.
- Technical specifications regarding the various measurement systems have been added in chapter 2.

**Specific comments**

Line 23-25: Statement is too general. For example in foggy conditions or in the absence of aerosol other instruments are preferable.

**Response:**
Statement was rephrased.

Line 27: "Emitted laser beam" would be more precise than "reference".

**Response:**
Corrected

Line 28: Pearson (2009) and subsequent citations therein might be a better references here, because they relate to the instrument used in this study. (Pearson, G., Davies, F., & Collier, C. (2009). An Analysis of the Performance of the UFAM Pulsed Doppler Lidar for Observing the Boundary Layer, Journal of Atmospheric and Oceanic Technology, 26(2), 240-250).
**Response:**
The reference is indeed very relevant and was added. Thank you

Line 28-30: Statement is too general. I recommend listing specific advantages over a particular measurement technique.

**Response:**
Corrected as required.

Line 31-34: The provided examples should be supported with references to literature.

**Response:**
Corrected as required.

Line 39: More importantly, the mathematical manipulations require certain assumptions on the state of the atmospheric flow.

**Response:**
Corrected as required.

Line 43: Please provide references for the various studies mentioned.

**Response:**
The sentence was removed. The studies are presented in the table..

Line 45-47: The following study also validates this lidar type against a tower: Newsom, R. K., Brewer, W. A., Wilczak, J. M., Wolfe, D. E., Oncley, S. P., and Lundquist, J. K.: Validating precision estimates in horizontal wind measurements from a Doppler lidar, Atmos. Meas. Tech., 10, 1229–1240, https://doi.org/10.5194/amt-10-1229-2017, 2017.

**Response:**
The table was updated.

Line 83: The site description could be more precise. What is the topography surrounding the site? Are there surface roughness elements like plants, trees or buildings and what is their approximate height (important to know if the measurements might be affected by the roughness sublayer)?

**Response:**
A detailed description of the location including land use and topography is added.

Line 105: Specify which frequency that is (e.g. the laser-repetition frequency or the sampling frequency of the return signal).

**Response:**
Sampling frequency , now  presented in table 2.

Section 2.2, 2.3 and 2.4: Information provided for the instruments other than the lidar is thin. What is the accuracy / precision for those instruments? Did the tethered balloon record its position and drift? Why the random sampling time for the tethered balloon? On which site of the tower were the instruments located (tower effects)? How often were the radiosondes launched?

**Response:**
- Accuracy and precision of the instruments is now reported.
- The tethered balloon did not record its position, however considering an operational  wind speed threshold of  12 m/s and the operation heights not higher than 1000m the horizontal drift is no more then 150m.
- The variables sampling time of the tethersondes results from radio connection issues.
- Exact locations of anemometers mounted on the mast is described in section 2.3
- Radiosonde launch timing is referred to.

Line 157-159: How many pulses were averaged for one estimate of the radial velocity?

**Response:**
10,000 pulses per ray. Now mentioned  (l.184).

Line 213-214: Doppler lidars are also able to observe the turbulence state from the variance / standard deviation of the velocity, which can provide information on atmospheric layers, too.

**Response:**

This is true, and would be an interesting direction for future studies. In this study the reference instruments did not allow turbulence measurements, and therefore, we limited ourselves to Lidar observations of the average wind only.

Line 230: Is there a significant trend between the average wind direction difference and the wind speed?

**Response:**
As figure 3 reveals, for strong winds of about 9-10 ms$^{-1}$, the direction difference is somewhat larger. Similar trend is evident also for the comparison with the mast (figure 4). The reason is not clear to us, and we hope to address this issue in future campaigns.

Fig 3a: I counted something around 67 data points, which is far less than I would expect for a three-day period. Is the difference explained by the filtering criteria of the LiDAR SNR and the height deviation of the tethered balloon? If yes, then that should be explained in the text and stated how much data was rejected. Also, the variable names and formatting of minus sign in equation of the linear regression could be improved.

**Response:**
The number of profiles used for each regression was added table 3 ,the batteries of the tethered balloon system had to be replaced every 6-7 hours. This means offline time for replacement + "bad" data from descent and ascent of the balloon (~74 points of averaged data , 9-12 lidar profiles for each averaging)

Caption of Figure 3: Unless I missed it earlier, this is the first time that the duration of the measurement campaign is mentioned. That information should be provided in the main text at the beginning of the results or in the methods.

**response**:
Dates are now mentioned in the beginning of section 2.

Line 240-241: Is there also a difference between the two height levels? One would also expect to see larger differences closer to the surface. If the authors can extract stability information from the measurements, the dependency of the errors to the stability could be also interesting (as it affects the horizontal homogeneity).

**Response:**
- Difference between the two heights was checked and found insignificant (R^2 of 0.94 for the low boom and 0.95 for the high boom).
- Differences between stabilities yielded R^2 0.95 for daytime and R^2 0.83 for nighttime. Although one would expect night time to yield better results (due to lower noise and better lateral homogeneity), our results show the opposite. The reason is that the wind speed vertical gradient in stable (night time) conditions in the few hundred meters close to the ground is much stronger than during neutral or unstable conditions. Since the Lidar

spatially convolves over the pulse length, error is accumulated when comparing to an in situ instrument.

Figure 4a: As for the previous figure, the number of the data points is not clear to me. Is the figure showing both tower booms together? Has there been some filtering criteria applied to the data?

As it is, I believe there are not enough data points for both booms and too many for a single boom.

**Response:**
The mast comparison shows both booms.The number of averaged profiles was  96  (9-12 lidar profiles for each averaging period)  resulting in 192 points for both booms.

Figure 5 and 6: It should be clarified if the SNR threshold from the methods section applied to those figures.

**Response:**
The caption of figure 5 addresses this issue

Line 350: While wind direction seems to agree, the wind speed is overestimated quite a lot by the model (more than a factor of two at times it seems). Therefore, I recommend providing objective error values instead of using satisfactorily to describe the agreement.

**Response:**
Figures 10 and 11 (in the revised ms) were modified to reflect wind direction and speed differences.

Figure 9 and 10: It might improve clarity of the figures, if the wind speed and wind direction information is separated into two panels.

**Response:**
Figures 10 and 11 (in the revised ms) were modified to reflect wind direction and speed differences.

Figure 11: Indicating the times of sunrise and sunset would help following the discussion.

**Response:**
sunrise and sunset times were added to all relevant figures

Line 407-408: Sentence should specifiy, that the preference of 60° over 80° is for the extraction of wind direction and wind speed (because it might be different, if one would extract other quantities from the measurements).

**Response:**
This is explicitly mentioned in the revised manuscript.

Also, the limitations of the study should be highlighted. Only one site is considered here and the conditions only covered what is considered a radiation driven diurnal cycle of the atmospheric boundary layer.

**Response:**
The new revision highlights the limitation in the introduction and in the discussion chapters.

**Technical corrections**

Line 10: Capitalize "D" in Doppler. And I recommend checking the journal guidelines for the capitalization "Lidar".

**Response:**
corrected

Line 45/46: Inconsistent spelling of "Stream Line".

**Response:**
Corrected

Line 68 and 70: Consider using "horizontal" instead of "lateral". Lateral is some fields used to specify a velocity component perpendicular to the streamwise direction.

**Response:**
Corrected

Line 74: I believe the abbreviation WRF was not introduced.

**Response:**
Added in the abstract.

Line 90: Dayan and Ednizik (1999) are also a good reference here. Dayan U, Rodnizki J (1999): The temporal behavior of the atmospheric boundary layer in israel. J. Appl. Meteorol. 38(6):830–836

**Response:**
The reference was added.

Figure 1: I might be wrong about this, but I believe the acknowledgment policies of Google also require the acknowledgment within the figure itself.

**Response:**
Corrected

Line 96: I recommend spelling out "four" here (similar instances throughout the text later on e.g. line 123, ).

**Response:**
Corrected

Table 2: Check units and number formatting throughout the table. Also, is the angle range reported for the zenith correct?

**Response:**
Units were organized better.
The elevation angle is correct - the manufacturer  meant that the lidar is able to scan 180 degrees in elevation   +3 degrees under horizon in both limits of the scan (this is a physical ability). corrected to the range -3-90 degrees.

Line 125: Spaces between number and unit.

**Response:**
Corrected

Line 143: Double citation.

**Response:**
Corrected

Line 147: Remove brackets from citations that are embedded into the sentence (also in line 166, 177 etc.).

**Response:**

Corrected

Line 151 and 211: Formatting of the subsection heading.

**Response:**
corrected
Line 184: Remove "out".

**Response:**
Corrected

Caption of Figure 3: The beginning of the figure caption is usually capitalized.

corrected
**Response:**
Line 239 and 246: The "s" in "m/s" should be written with a power to "-1".

corrected
**Response:**
Line 246-247: Inconsistent use of spaces.

**Response:**
Corrected

Line 254: As it is a continued discussion of Table 2, I believe a new paragraph is not needed here.

**Response:**
corrected
Figure 6 and 7: The formatting of dates is inconsistent.

**Response:**
corrected
Line 404: Replace "Out" with "Our".

**Response:**
corrected